# Opponent processes in visual memories: A model of attraction and repulsion in navigating insects' mushroom bodies

**Florent Le Möel** [iD]**, Antoine Wystrach** [iD]*

Research Centre on Animal Cognition, University Paul Sabatier/CNRS, Toulouse, France

* antoine.wystrach@univ-tlse3.fr

**Data Availability Statement:** Code is available on GitHub (https://github.com/antnavigationteam/opponent_process_ploscompbiol2020).

## Abstract

Solitary foraging insects display stunning navigational behaviours in visually complex natural environments. Current literature assumes that these insects are mostly driven by attractive visual memories, which are learnt when the insect's gaze is precisely oriented toward the goal direction, typically along its familiar route or towards its nest. That way, an insect could return home by simply moving in the direction that appears most familiar. Here we show using virtual reconstructions of natural environments that this principle suffers from fundamental drawbacks, notably, a given view of the world does not provide information about whether the agent should turn or not to reach its goal. We propose a simple model where the agent continuously compares its current view with both goal and anti-goal visual memories, which are treated as attractive and repulsive respectively. We show that this strategy effectively results in an opponent process, albeit not at the perceptual level–such as those proposed for colour vision or polarisation detection–but at the level of the environmental space. This opponent process results in a signal that strongly correlates with the angular error of the current body orientation so that a single view of the world now suffices to indicate whether the agent should turn or not. By incorporating this principle into a simple agent navigating in reconstructed natural environments, we show that it overcomes the usual shortcomings and produces a step-increase in navigation effectiveness and robustness. Our findings provide a functional explanation to recent behavioural observations in ants and why and how so-called aversive and appetitive memories must be combined. We propose a likely neural implementation based on insects' mushroom bodies' circuitry that produces behavioural and neural predictions contrasting with previous models.

## Author summary

Insects such as ants and bees are excellent navigators, able to learn long foraging routes and return to their nest in complex natural habitats. To achieve this, it is believed that individuals memorise views–the visual scene as they perceive it–only when their body is precisely oriented towards the goal. As a result, the insect can return to its goal by simply being attracted in the direction that represents the highest visual familiarity. Here we use a

**Funding:** This study was funded by the European Research Council: ERCstg EMERG-ANT 759817 attributed to AW. The funders had no role in study design, data collection and analysis, decision to publish, or preparation of the manuscript.

**Competing interests:** The authors have declared that no competing interests exist.

computational approach to show that this strategy suffers from a major weakness: a single view of the world does not suffice to tell whether the agent should turn or not to reach its goal. However, a surprisingly robust solution to this problem arises if we simply assume that these insects memorise not only goal-oriented views but also anti-goal-oriented views that they then treat as repulsive. This idea clarifies several observed behaviours that were difficult to explain with previous models. Overall, this research helps us to understand how insects combine memories in specific brain areas and can navigate so efficiently despite their tiny brain.

## Introduction

Early naturalists were impressed by the navigational abilities of central place foragers such as bees, wasps and ants [1,2] and notably, from the comprehension that these abilities arose from a tiny volume of brain matter [3]. What mechanisms underlie insect navigation are still far from clear, however our understanding has considerably improved in the last decades. By combining experimental and computational approaches (e.g., [4]), this field has explored, tested and refined parsimonious hypotheses about insect navigational mechanisms [5]. Nowadays, advances in insect neurobiology [6,7] enable the design of biologically-realistic neural models to explore how such navigational mechanisms may be implemented in the insect brain [8–11]. The current work uses a computational approach to explore a novel hypothesis on the use and storage of visual memories for navigation. We chose to model an ants' perspective; however, we believe our conclusions can easily be extrapolated to other insect navigators.

### Ant visual navigation and modelling

Naive ant foragers control their first journeys by relying heavily on Path Integration [12–14] or social cues [15,16]. These early-available solutions provide them with the opportunity to learn and memorise visual scenes for their subsequent journeys. They then predominantly rely upon these memories once experienced [17–19].

Modelling studies have proposed several types of algorithms to explain how insect memorise and use views for guidance. While most models assume that guidance is achieved by comparison of the currently perceived view to a memorised view (although this is still debated in bees [20,21]), they vary greatly on the way this is implemented (see [22] for a comprehensive classification). For instance, some homing models need to store only one view at the goal acting as an attractor [4,23] while others require multiple memorised views acting as direction setters pointing along a learnt route [18,24,25] or towards the goal [26–28]. Also, some models compare directly the raw visual input [29–32] while others are based on some specific visual processing such as distorting the current view [26], performing Fourier transformations [33,34], or extracting and recognising specific features from the environment [4].

### Familiarity-based models

In the last decade, a particular type of models, which we call here 'familiarity-based' has gained attraction for it provides good explanations of a variety of observed ants' navigational behaviours [18,24,27,28], and is consistent with the insect's neural circuitry [9,29].

A key to these familiarity-based-models is that insects memorise visual scenes in an egocentric way, so that recognition is dependent on the insect's current head direction [35]. In other words, a current view will appear familiar only when the insect is aligned in (roughly) the

same direction than it was at the time of learning. If the insect is located at a familiar location, but its body is oriented in a wrong direction, its current view will appear unfamiliar. Perhaps counterintuitively, this direct coupling between visual familiarity and head direction can provide a solution for navigation. During learning, all one needs to do is to memorise views while facing a relevant direction, such as the direction of the nest as indicated by the Path Integrator [12,14]. The agent can therefore later seek the direction of the goal by simply looking for the most familiar direction of the current location. This strategy enables to recover a goal direction over tens of metres in complex outdoor environments [36,37] and, interestingly, is most robust when the scene is parsed at the low resolution of insect eyes [31,33].

This principle can be directly applied to recapitulate idiosyncratic routes, given a set of visual memories stored along them. Agent-based simulations have shown that the 'walking forward in the most familiar direction' rule enables agents to recapitulate routes [8,24,29,38] in a way that closely resembles what is observed in solitary foraging ants [18,39,40]. These models are further supported by the well-choreographed 'learning walks' displayed by ants–as well as the equivalent 'learning flights' displayed by bees and wasps [41–45]–around their nest (or a food source). During learning walks, ants appear to systematically pause while facing towards the nest (or a food source), presumably to memorise 'correctly aligned' views from multiple locations around the goal [12,14,46–48]. Given such a visual memory bank, models show that the exact same principle used for route following is able to produce a search centred on the goal location [24,27]. There again, the models' emerging search patterns capture several aspects of the ants' visually driven search [38,46,49,50].

The greatest appeal of familiarity-based models is that the comparison of the current and memorised view needs only to output one familiarity value. To get this familiarity value, there is no need for finding correspondence between parts of the visual field and a direction in the world. This contrasts with so called 'correspondence models' where features must be extracted from panoramas, and matched locally. Familiarity-based models do not need retinotopic mapping to be preserved at the comparison stage. This fits well the connectivity of the insect Mushroom Bodies [51], where memories of visual scenes are likely encoded [8,9]. Indeed, in visually navigating insects, the visual input as perceived through their low-resolution eyes and processed by early optic neural relays [52–54], is then projected to more than 100,000 Mushroom Body cells (Kenyon Cells: KCs) through an apparently random pattern of connections, likely disrupting retinotopic organisation [55–57]. Remarkably, given the synaptic plasticity observed at the output of the MB during learning [58–60], the resulting activity of the MB output neurons naturally provides a measure of familiarity of the current sensory input [8,51].

A simplified model of the ant and bees circuitry shows that the Mushroom Bodies provides enough 'memory space' for an agent to store the required visual information to recapitulate a 10 meter long-route in a naturalistic environment–here again, by simply following the rule of 'walking forward in the most familiar direction' [8].

## Limits of familiarity-based models

Familiarity-based models have the advantage of capturing a wide range of observed behaviours (route following, nest search, the need for learning walks and scanning multiple directions when lost) while remaining remarkably simple and faithful to the known insect circuitry. However, they suffer from one main drawback: the familiarity of the view currently perceived does not tell whether the current direction is right or wrong. For instance, a view perceived when facing away from the nest can still provide a good match with a view acquired during learning walks (like when the insect was located on the other side of the nest), and thus appear highly familiar, although the current direction is dead wrong. Conversely, a view perceived

when facing towards the nest but several meters away from it may provide a rather poor match, although the current direction is correct. In other words, current familiarity does not correlate with directional error. Therefore, it is problematic for these models to infer a correct motor decision uniquely from the current visual familiarity.

Typically, familiarity-based models eschew this problem by using a 'stop-scan-go' strategy: at each steps the agent stops and rotates on the spot to scan multiple directions; and subsequently moving in the direction that is most familiar for this particular location [8,24,26,27]. However, this procedure is (1) cumbersome, as it requires the agent to stop, rotate on the spot (i.e. to trigger a scan) and compute the familiarity for many directions at each step; (2) non-parsimonious, as it requires a three-stage process (i.e., scan/select a direction/step forward) and the assumption that the agent repeatedly stores and compares the familiarities across all sampled directions at each step, to select the highest one; (3) unrealistic, as ants and bees do not need to scan in order to home successfully.

A substitute to physically rotating on the spot could be to perform some sort of mental rotation [34]. However, it is yet difficult to see how mental rotation could be implemented in the insect brain. Such a feat would presumably require that visual encoding is achieved on the 360 visual field at once (as in [34]), but insects visual neuropils in each hemisphere seem to receive almost purely projections from the ipsilateral eyes, which visual field covers around 180 degrees only. Also, an ability for mental rotation seems hard to conciliate with the extensive physical scanning behaviours displayed by ants during a moment of visual uncertainty [61]. We decided here to remain entirely faithful to the known insect neural connectivity.

An alternative to performing rotations (either physically or mentally)–based on lateral oscillations–has been proposed for route following[25], but as we show here proves to be extremely sensitive to parameter change. This is expected as such an alternative does not solve the intrinsic problem that current familiarity does not correlate with directional error.

## Our approach

We present here a simple and realistic solution the problem of previous familiarity-based models. We assumed that navigating insects store two separate memory banks in their Mushroom Bodies: one attractive–based on views learnt while facing the goal direction–and one repulsive–based on views learnt while facing the anti-goal direction, that is, 180˚ away from the goal direction; and that the output of both memory pathways are integrated downstream during navigation (Fig 1). These assumptions are sensible because they flow from behavioural and neurobiological observations rather than the desire to improve a model. First, ants of the genus *Myrmecia* were shown to display regular alternations between phases while facing the nest and when facing the opposite direction, suggesting that they may learn both attractive and repulsive visual memories [48]. Second, ants can associate views with an aversive valence, triggering turns when these views are subsequently experienced [62]. in insects' brains, it is now clear that several output neurons from the Mushroom Bodies (so-called MBONs), which typically respond to learnt stimuli, convey either attraction or avoidance signals [58,63–65]; and that these MBON signals with opposite valences are integrated downstream [66–68].

We explored the effect of these assumptions using views taken from virtual reconstructions of ants' natural outdoor environments [69]. Our results show that such an integration of visual memories is akin to an opponent process, and provides a qualitative gain in navigational information. Notably, the integrated signal now directly correlates with the angular error of the current direction and thus greatly facilitates the problem of steering. As a proof of concept, we implemented this opponent memory principle in a simplistic ant agent model [25]. This granted the agent the ability to home with much higher success, and drastically increased

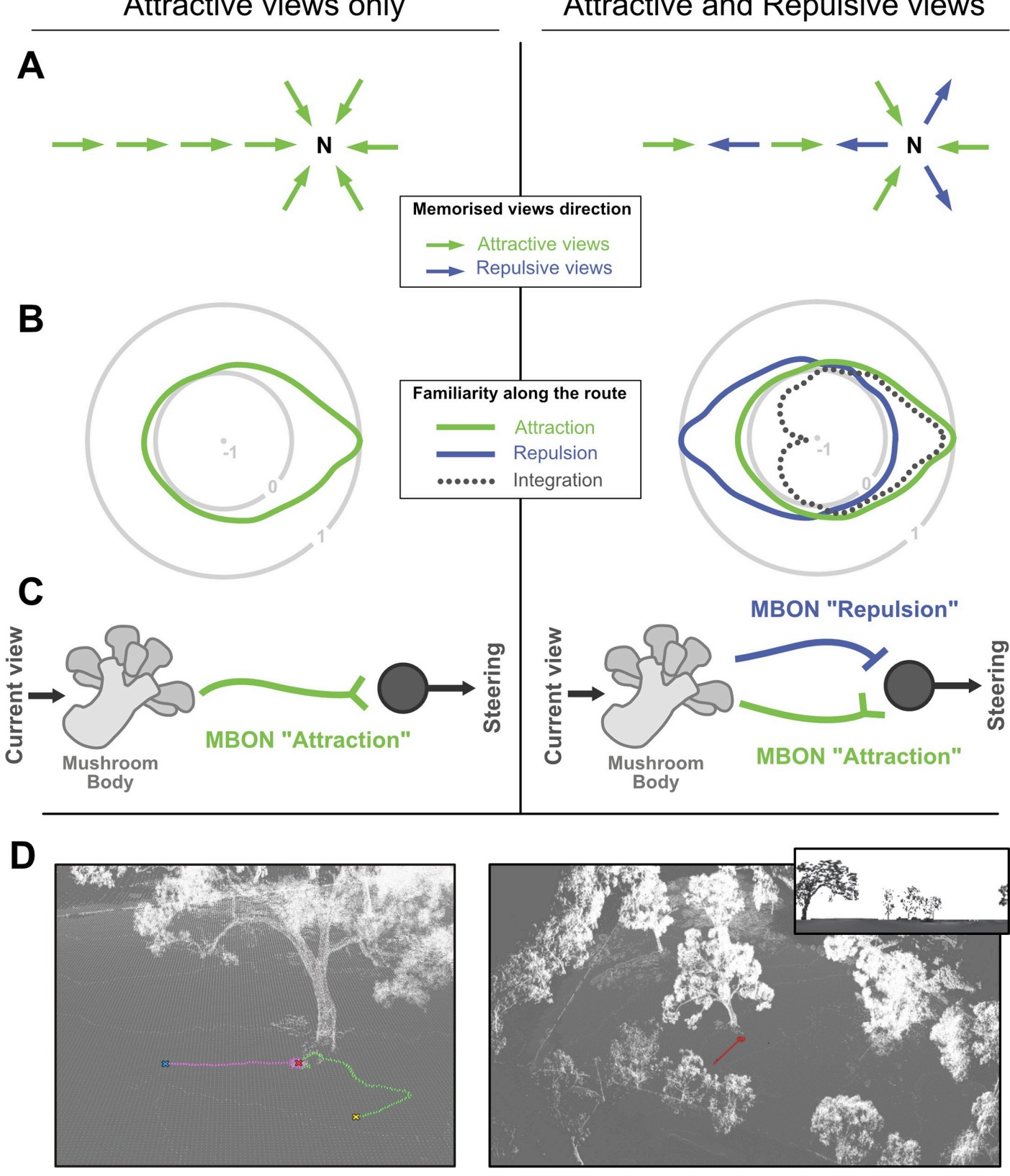

**Fig 1. Model structure and core concept. A.** Schematic representation of the spatial distribution of memorised views, assuming here the agent has learnt views around the nest and along a homing route. Left, only views with a positive valence (i.e. attractive) are learnt, all pointing towards the nest (N). Right, views with positive valence as well as views with negative valence (i.e. repulsive) are learnt. Repulsive views are directed 180° away from the nest. **B.** Schematic representation of the visual familiarity perceived across all 360° directions for a given position: here on the familiar route with the nest direction towards the right. Concentric circles represent the level of familiarity of the current view (i.e., how well it matches the memorised view) as a function of the direction of the current view: 0 indicates no match whereas 1 is a perfect match. Given the opponent integration of both memory pathway (dash line), negative values indicate directions for which the current view matches the repulsive memories better than the attractive ones. **C.** Scheme of the proposed neural implementation; the Mushroom Body receives information about the current view. Learning consist in modulating the connections elicited by goal oriented views with one Mushroom Body output neuron (MBON). When subsequent views are presented, the signal of the MBON thus correlate with the current visual familiarity (see [9] for more detail). With the opponent process (right), learning additionally involves anti-goal oriented views to be associated with another MBON conveying the opposite valence. Both MBONs are integrated downstream by simple subtraction before modulating the steering. **D.** Example snapshots of the virtual 3D worlds used in our navigation simulations. Coloured lines are paths of the agent. In inset, an example of what the model gets as visual input (horizontal resolution of 5°/pixel).

robustness to both displacement and parameter changes. Our hypothesis makes clear behavioural and neurobiological predictions, as well as calls for further hypotheses which we discuss.

## Results and discussion

The performance of insect visual navigators is typically thought to be based on visual memories acquired while facing in the correct goal direction [8,24–29,35,36,38,70–72]. This is strongly corroborated by the learning walks and learning flights of ants [12,14,46,47], bees [41,44,45] and wasps [42,43], where the individuals regularly turn back and look towards the goal direction, thus storing the information necessary to subsequently return to it. Indeed, with such a memory bank of goal-oriented views, the most familiar direction of each location tends to be directed along the route or towards the nest, resulting in 'familiarity flows' across space converging towards the goal (Fig 2A, left; as in [30–32,34]). Thus, all the insect needs to do to reach its goal is, at each location, to walk forward in the direction that presents the most familiar view [24,27,28].

### The drawbacks of using only goal-oriented views

The problem of using only 'goal-oriented' visual memories becomes apparent when mapping the familiarity across multiple locations when always facing in a same direction (Fig 2B): with goal-oriented views only, familiarity increases as one gets closer to a location where a view facing in that same direction had been memorised (Fig 2B, left). As a result, the familiarity of a given view tends to correlate with distance from the nest but not with the angular difference from the nest direction (Fig 2C, left). That is to say, without scanning multiple orientations, familiarity does not reliably indicate whether one should turn or carry on straight ahead. For instance, we show that a view that is correctly oriented towards the goal can present the same familiarity as a view that is poorly oriented (180° opposite) from the goal (black and white squares in Fig 2B, left). To summarise, when using goal-oriented views only, one can infer the correct direction by scanning multiple directions at a given location (Fig 2A, left), but the familiarity of a given view, taken alone, is not informative about the current directional error and thus cannot be used directly for steering (Fig 2B, left).

### Opponent processes in visual memories

We show here that the need for scanning can vanish entirely when assuming the co-existence of a second memory pathway, based on views learnt this time while facing in the anti-goal direction. This idea is corroborated by the recent observation that *Myrmecia* ants during their learning walks display oscillatory alternations between nest-facing and anti-nest-facing phases [48]. The key is that the familiarity signal outputted by the anti-goal memory pathway must be

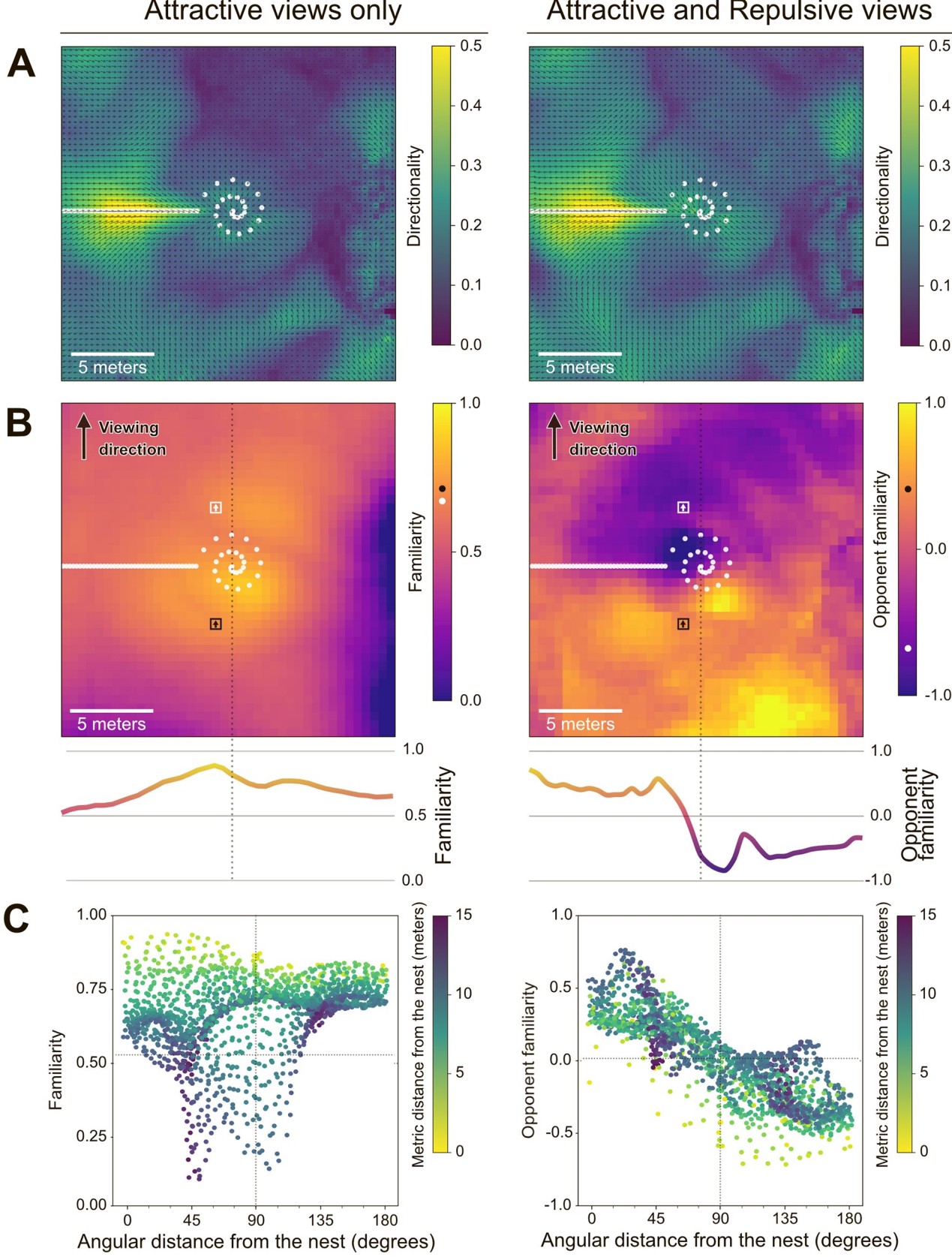

**Fig 2. The statistics of opponent visual familiarity across space. A-C.** Left column, Attractive views only; Right column, Attractive and Repulsive views. **A.** Top view spatial map of the directional specificity of the visual familiarity, on a 30 x 30 metres area centred on the nest given view stored around the nest and along a homing route (white dots). Arrows indicate the direction presenting the maximum familiarity for each position on the map. Colour and arrows lengths indicate the specificity of the best matching direction as compared to the other directions at this location (calculated as 'max familiarity'–'median familiarity'), normalised between 0.0 (least specific) and 0.5 (most specific). **B.** Visual familiarity across the same 30 x 30 area as in A, but given one facing direction only (North). Although this direction (North-facing) was picked rather arbitrarily, any other direction provided qualitatively similar results, with one exception: route-directed views. We deliberately excluded this direction, to show the apparent familiarity as seen from unfamiliar viewing directions. Familiarity values (colours) represent the match between the north facing view at this location and the memory bank (the view that gives the smallest difference), normalised between 1.0 (i.e., perfect match) and 0.0 (worst match). On the right panel, 'Opponent familiarity values' show the integration of both memory pathways, with negative values indicating that the view at this location matches better the repulsive memory bank. Two positions located on opposite sides of the nest (black and white squares) may appear similarly familiar (black and white dots on the colour map) given Attractive memory bank only (left panel) but show opposite valence given Attractive and Repulsive integration (right panel). The curves in the bottom panels, shows the familiarity along a transect passing through the nest (dash line). **C.** Scatter plot of the familiarity values of B, against the angular distance between the north facing view and the nest direction. Angular distance of 0° means that the north facing view is pointing towards the nest. Metric (Euclidean) distance of the position of the view from the nest is shown in the colour map.

subtracted from the signal of the goal-oriented memory pathway (or the other way around). In other words, goal-oriented and anti-goal oriented memories pathways have opposite valence and can be considered as driving attraction and repulsion respectively. The signal resulting from this integration can be seen as determining whether the current view is attractive or repulsive.

This provides several benefits. First, at a given location, recovering the correct direction through scanning is facilitated because anti-goal memories tend to make anti-goal directions all the more repulsive, which thus increases the difference between incorrect and correct directions (Fig 1B, right). Second and most importantly, a single view is informative: across locations, views oriented towards the nest appear attractive, while views oriented away from the nest appear repulsive (Fig 1B, right). In other words, the 'opponent familiarity' resulting from such an integration is not correlated with distance from the goal but with the current directional error (Fig 1C, right) and can thus be used directly for steering (Fig 3).

The concept of integrating goal and anti-goal visual memory pathways is to some extent analogous to an opponent processes, albeit not at the perceptual level–such as what has been proposed for colour vision in primates [73] and bees [74] or polarisation vision in invertebrates [75–77]–but at the level of the environmental space. In the case of polarisation vision, R7/R8 polarisation-sensitive photoreceptors have orthogonal orientation: opponent polarisation neurons receive excitatory input by one direction of polarisation and inhibitory input from the orthogonal direction [75]. The key advantage of such process is that it provides consistency against variations of luminance levels. For instance, if the intensity of light decreases, both excitatory and inhibitory inputs decrease, but their ratio stays alike.

In much the same way, we propose here that 'opponent familiarity neurons' receive excitatory input from the familiarity signal based on the memories stored in one direction (e.g., goal oriented view) and inhibitory input from the familiarity signal based on memories stored in the opposite direction (e.g., anti-goal oriented view) (Fig 1C). The resulting signal provides consistency over overall variations in familiarity across locations. For instance, as one is getting closer to a familiar location, both inhibitory and excitatory familiarity pathways decrease but the sign of their difference stays alike. As a result, the opponent familiarity obtained is quite insensitive to variation in distance from the locations at which memories have been stored, but correlates remarkably well with the current directional error (Fig 2C, right). Obviously, in a completely unfamiliar environment, both memory pathways output similarly low familiarity values and the resulting opponent familiarity remains null whatever the direction currently faced.

Absolute mismatch values resulting from the comparison of two views are dependent on the structure of the scenes. Locations presenting rather homogenous visual patterns across

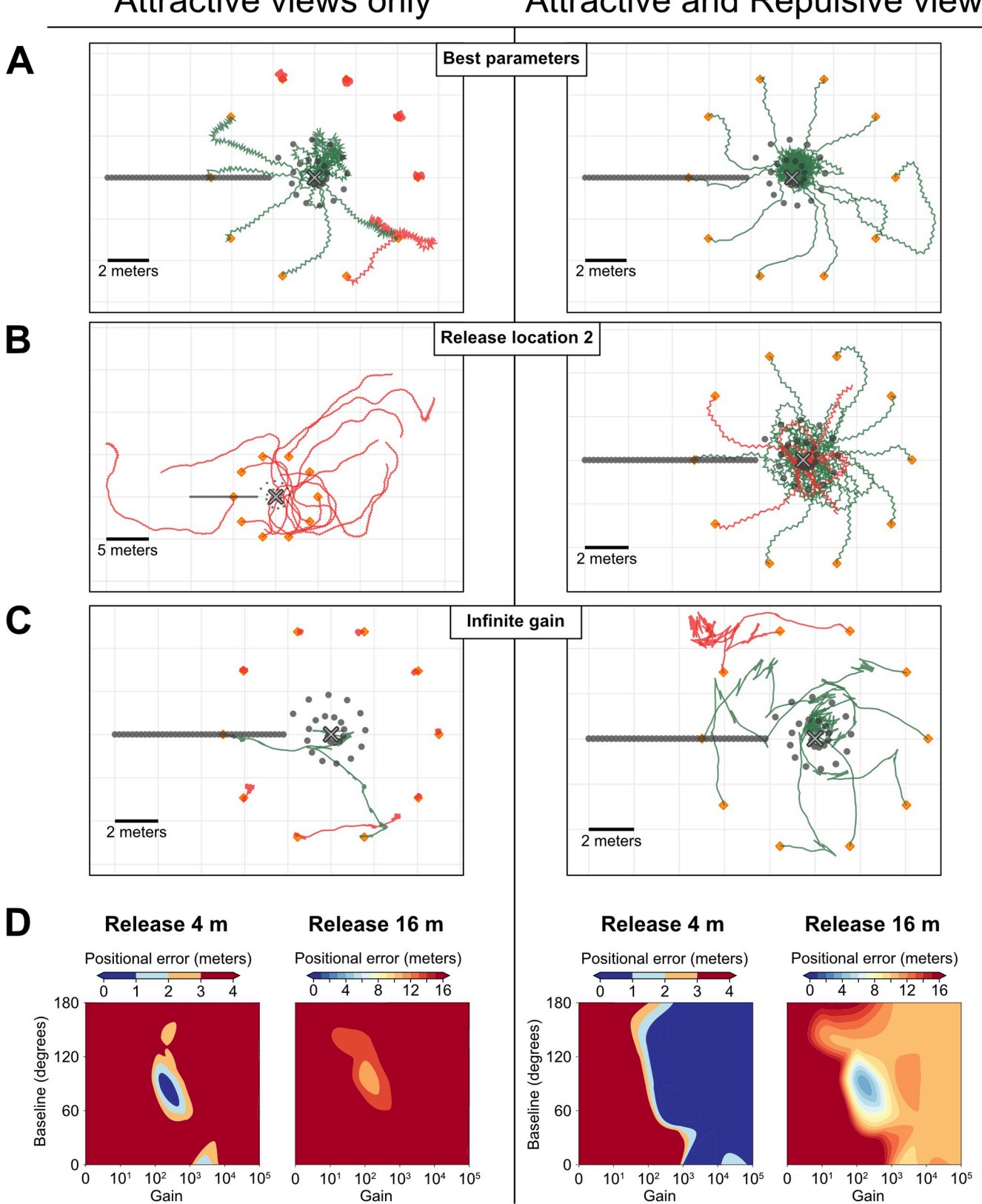

**Fig 3. Efficiency in finding the nest of homing agents. A-D.** Comparison of oscillatory agents implemented with attractive views only (left column) or using the opponent integration of attractive and repulsive views (right column). **A-C.** Example paths (120 steps) of 10 individual agents for each condition. Grey cross, nest; Grey dots, positions of the learnt views (facing towards the nest); Orange markers, release locations of the agents. Green paths correspond to agents that successfully ended up within 1-meter radius of the nest, red paths correspond to agents that failed to do so. **A.** Paths given the best set of parameters obtained for the 'Attractive-only agent' at this nest location. Note that as a result of the poor success of this model, agents staying at the starting point are favoured over agents straying further away. **B.** Paths of agent trained and tested at a second nest location in the virutal world, but using the parameter optimised for the location in **A**. Agents using Attractive view only appear less robust to changes in the environment. **C.** Example of a simulation with an infinite gain parameter, which constrains the model to 0˚ or 180˚ turns. Thanks to the addition of ± 20˚ random noise on turning direction, the Attractive-Repulsive model can still home. **D.** Heatmaps representing homing success (median of 10 agents' arrival distance from the nest) across variations of two parameters. The 'baseline' indicates the turn angle effected in response to an average familiarity value (or when attractive familiarity = repulsive familiarity). The 'gain' is a value that convert the perceived familiarity into the turn modulation, which increases (if current attractive familiarity is low) or decreases (if current attractive familiarity is high) the turn amplitude in relation to the baseline. A high gain makes the agent very reactive to small change in familiarity. Agents using attractive view only requires a fine parameter tuning to home successfully at 4-meters, and no set of parameters allow for significant homing abilities from 16 metres or more. Agents released at 4m using the opponent integration of attractive and repulsive memories are robust as long as gain is above a threshold value, but parameter tuning can improve their performance from 16m.

azimuths tend to yield low mismatches, while heterogonous scenes tend to yield high mismatches. For instance, if an agent ends up unusually close to a tree, the mismatch experienced may become unusually high, that is, the familiarity becomes unusually low (see for instance Fig 2B, left). Thus, what should be considered as a good or bad familiarity value is a tricky question, and this is why models using only attractive view without scanning are very sensitive to parameter calibration (Fig 3). This problem disappears when using an opponent process since the resulting 'opponent familiarity' is always relative to the two measurements: determining which of the two signals is stronger tells whether one is well oriented or not. In other words, instead of knowing 'how good is the current familiarity', the navigator assesses if 'the current view matches better with a good or a bad direction'. As a result, a glance in a single direction is sufficient to inform the decision to turn, or rather go forward.

## Opponent visual memories enable robust navigation

To test whether such an opponent process in visual memories is actually helpful for navigation, we implemented it in a simple agent navigating in reconstructions of ants' natural environments (Fig 1D, S2 Fig). The goal of the following section is not to provide an exhaustive exploration of such a system, but to serve as a proof of concept that implementing two opponent memory banks in a navigating agent can be quite useful and straightforward. We chose a previously published simple agent that has the advantage to estimate the familiarity of a single view per step, rather than scanning multiple directions [25]. In this model, guidance involves an oscillator resulting in a continuous alternation between left and right turns, as observed in several insects, including ants [78–80]. The amplitude of the oscillations is simply modulated by the familiarity of the view currently perceived: familiar views trigger small turns, whereas unfamiliar views trigger large turns; however, the direction (left or right) of the turns is solely dependent on the current state of the oscillator. When using goal-facing memories only, this model can be sufficient for an agent to recapitulate a route in a natural-like environment [25]. This is because, as long as the agent stays on the familiar route, visual familiarity is higher when the currently faced direction is aligned with the correct route direction.

However, our parameters exploration shows that this 'attractive-views only' model is operational only given a small range of parameters (Fig 3D, left), and when operational, outputs mediocre homing performance (Fig 3A, left). Moreover, the range of parameters at which it operates in a given region of the world does not necessarily operates in other regions of the world (Fig 3B, left). Here again, this is because absolute mismatch values are strongly dependant on the visual structure of the scenes.

In contrast, when equipped with two memory banks acting as opponent processes, homing performance becomes much higher. Not only is the agent more accurate in searching at the goal location but homing becomes much more robust to variations in location (Fig 3A and 3B) and larger displacements, with successful homing from up to 16 metres in our Australian world given a learning walk spanning up to 2 meters around the nest (Fig 3D, S1 Fig). In other words, with the opponent process strategy, the agent is able to home robustly even when not inside a route corridor. Finally, and perhaps most importantly, homing becomes remarkably robust to changes in parameters (Fig 3D). Basically, it operates as long as the gain is above a given threshold value (Fig 3D). Briefly, 'gain' is a single-value parameter that converts a familiarity value into a turn amplitude, see Methods for details. A gain too small simply prevent the agent to execute sharp turns, and given that our agent is forced to step forward, it cannot reorient sharply enough to search at the goal. However, a gain too high seems to present no fundamental problem. An infinitely high gain makes the turn amplitude extremely sensitive to minute changes in familiarity perceived and thus constrains turn amplitude either to 180˚ (U-turn) when the view matches best the anti-nest memory bank or 0˚ (go straight ahead) when the view matches best the nest-oriented memory bank. Obviously in this binary condition (0˚ or 180˚ degree turn) the agent is doomed to be moving along straight line. However, providing a bit of random noise on the direction taken at each time step is sufficient to enable an agent with infinitely high gain to explore novel directions, and to eventually home successfully (Fig 3C).

The success of the agent is not due to the fact that it possesses twice as many memorised views than the previous agent with only nest-oriented memories. Given less than half the usual number of memorised views (10 of each, instead of 25 attractive-only), the agent using opponent process still homes well (Fig 4C). Furthermore, the position of nest and anti-nest oriented memories can be chosen quite arbitrarily (Fig 4A). Even though *Myrmecia crosslandi* ants appear to alternate regularly between nest and anti-nest facing directions [48], other species may not [61,81], and indeed according to our model, it is not needed. Also, as observed in ants, learning walks spanning larger areas provide larger catchment areas [82]. With our model, a learning walk spanning either a 0.5-metre radius or an 8-metre radius enables the agent to home reliably up to 8 and 16 metres respectively (S1 Fig), even though the number of memorised views remains the same. Interestingly, the agent still manages to pinpoint the goal given a learning walk spanning only a 10-centimetre radius if released in the vicinity (Fig 4E). Although, here the step length must be smaller than the usual 20 cm to ensure that the agent does not exit the small catchment area; after all, ants do tend to slow down when close to the goal [83]. Also, the model is readily able to navigate in a totally different virtual world such as the one reconstructed in the Spanish arid area of *Cataglyphis velox* ants [40] (Fig 4F), which presents much more local clutter and no distant landmarks (S2 Fig). Although here, because the views perceived change very rapidly with displacement, we needed to reduce the step length to prevent the agent to escape the familiar catchment area (Fig 4F).

Perhaps most interestingly, we observe that the memorised views need not be precisely oriented towards the goal or anti-goal direction. Homing is robust even to very large random deviations in memorised views orientation (e.g., ±90˚ around the goal and anti-goal directions) (Fig 4B). This may explain why ants displaying learning walks may not necessarily align their body precisely towards and away from the nest [12,48,61,81]. Finally, most of our simulations were achieved with a typical ant resolution of 5˚/pixel [53], but resolution does not need to be precisely tuned either. For instance, we observe that the agent still homes well given a resolution of 10˚/pixel (Fig 4D). Obviously, we expect that a too low or too high resolution would eventually disrupt the ability to home using such a visual matching strategy, as investigated elsewhere [31].

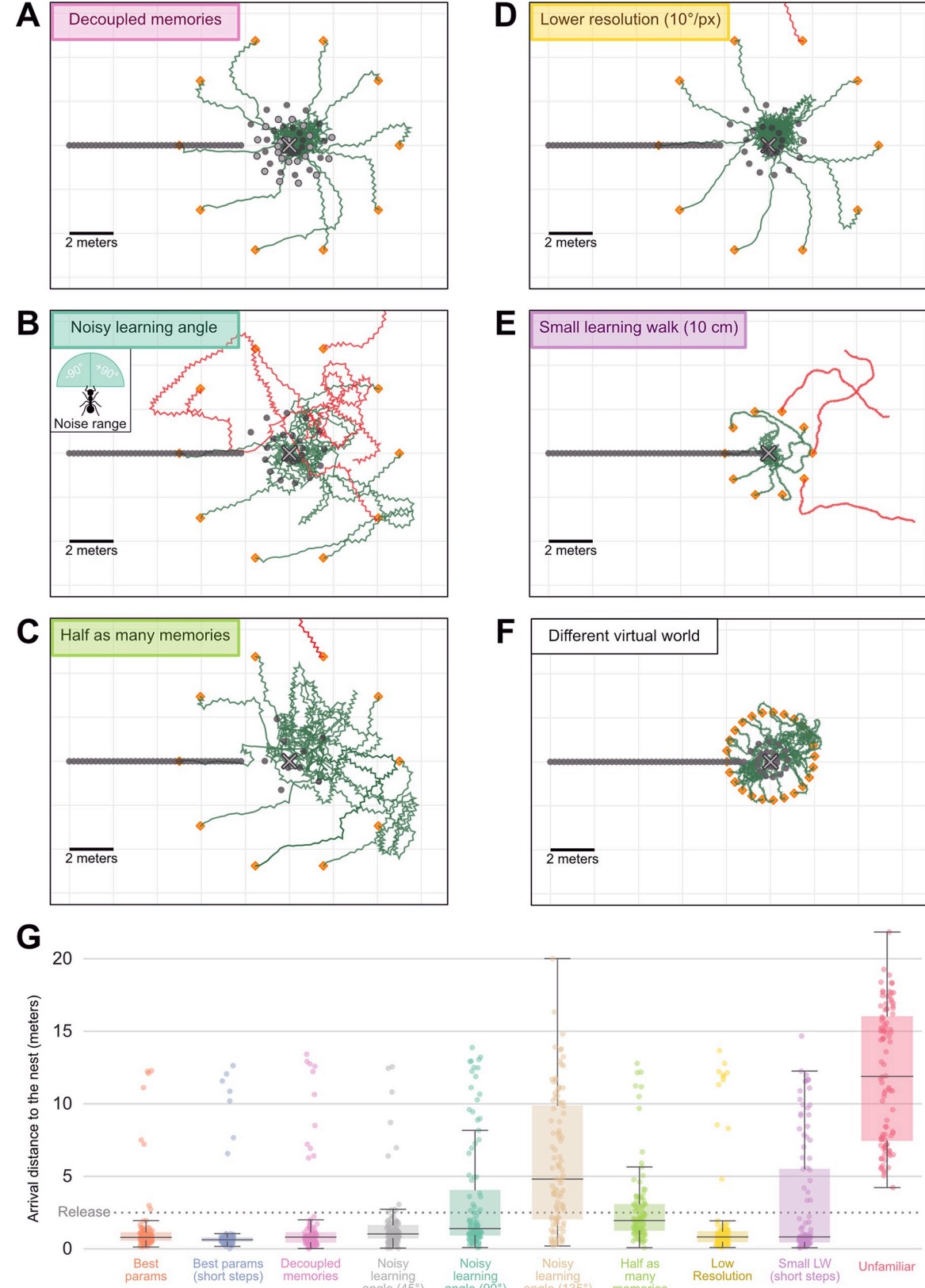

**Fig 4. Robustness of agents using attractive and repulsive views. A-F.** Paths (120 steps) of navigating agents using the opponent integration of attractive and repulsive views, in several experimental conditions. Symbols are the same as in Fig 3. **A.** Decoupled memories: attractive memories have been acquired at different locations (dark grey dots in a spiral) than the repulsive memories (light grey dots). **B.** Noisy learning angle: attractive and repulsive memories were acquired given a ±90° uniform noise in the viewing direction

around the goal and anti-goal direction. Inset, range of the noise. **C.** Half as many memories: only 10 learning walk views (instead of 25) were learnt. **D.** Smaller resolution: The visual input to the model is a coarser snapshot of the environment, with 10°/pixel (instead of 5°/pixel). **E.** Small learning walk: the learning walk along which memorised view are sampled span only a 10 centimetre radius around the nest (instead of 2 metres). The agent walked slower (4 times shorter steps, 4 times as many) in this condition. **F.** Test in a different virtual world, with more clutter and no distant landmarks (see S2 Fig). **G.** Boxplots of the arrival distance to the nest of agents released at 5m around the nest and walking for 30m (300 runs per conditions: 3 different nest locations in the virtual world x 100 release locations around each nest location). Conditions corresponds to the conditions mentioned above. 'Unfamiliar' corresponds to agents released around a fictive nest at an unfamiliar region in the virtual world.

It is important to note that we did not wish here to explore thoroughly how this particular agent responds to these different variables. We rather wanted to show that the principle of an opponent process in visual memories provides robust information which can be directly exploited for successful navigation.

## Neural implementation in the insect brain

The idea of using two memory pathways that interact antagonistically is directly inspired by insect neurobiology. Notably, recent work in *Drosophila melanogaster* has exposed the neural mechanisms underlying the formation and expression of aversive and appetitive memories in the Mushroom Bodies (MB) [63]. Stimuli identities are encoded as sparse activation patterns of specific Kenyon Cells (KCs). These KCs synapse onto multiple MB output neurons (MBONs), of which activities can, for example, favour an approach behaviour, or an avoidance behaviour [84,85]. Also, each MBON is associated with a dopaminergic neuron (DAN) conveying a specific reinforcement value. In a nutshell, during learning, the co-activation of a stimulus with a positive or negative reinforcer (DAN) decreases the stimulus elicited KCs' output synapses onto the associated appetitive or aversive MBONs respectively. Subsequently, this biases the stimulus-driven activity towards the residual pathway and thus favour either approach or avoidance behaviour [84,86].

Interestingly, these aversive and appetitive pathways are typically integrated in an antagonistic fashion, sometimes by having an 'aversive' MBON directly inhibiting a neighbour 'appetitive' MBON or by having MBONs with opposite valence conveying inhibitory and excitatory signals onto the same downstream neuron [63,66,67].

In fruit flies and honeybees, learning and memory involving the mushroom bodies is typically studied experimentally using olfactory stimuli [51,63,64,84,85,87–89]. Nonetheless, we observe abundant projections from visual areas to the MBs in visually navigating insects such as ants and bees [55,56,58,90,91] (and to a lesser extent, in *drosohpila* [92]), and neural models show that the identity of visual scenes can be represented in specific KCs activities and enable route navigation [8,9].

Therefore, in simple terms, the current idea assumes that guidance in insect visual navigator is driven continuously from a balance between attraction and aversion. Goal-oriented views are learnt as appetitive while anti-goal-oriented views are learnt as aversive by having their respective KCs view-driven activity directed towards antagonistic MBONs (Fig 1C). Both MBON pathways are integrated and the resulting signal is used for triggering approach (i.e., inhibit turn) or avoidance (i.e. trigger turn). As we have shown, this simple scheme enables robust navigation.

## Predictions and further hypotheses

The idea of using such opponent memory pathways for navigation yields predictions that contrast with previous models. Also, it enables us to formulate further hypotheses. We briefly state some of these below.

- Our model implies the activation of two different reinforcer signals to categorise whether views should be learnt as aversive or appetitive. One should be active while the ant is roughly facing in a specific direction (e.g., towards the goal), the other while the ant is facing in the opposite direction (e.g. away from the goal). One possibility is that these reinforcer neurons are under the control of the Path Integrator. This could be achieved by dopaminergic neurons conveying signals from the lateral accessory lobe (i.e. Central Complex output of path integration steering command) to the Mushroom Body lobes.

- Both appetitive and aversive pathway should convey antagonistic signals, which are likely implemented as inhibitory and excitatory synapses. The balance between attraction and aversion could thus be altered by pharmacologically by inoculating specific receptor antagonists (such as GABA receptor antagonist) in the output region of the MBONs.

- For ants at the exact nest location, visual familiarity should be very high for both opponent pathways, resulting in a balanced activity. In completely unfamiliar locations, familiarity is very low for both opponent pathways, resulting also in a balanced activity. This should be true for all directions at both of these locations. Therefore, the current model makes the counter-intuitive prediction that a similar behaviour is expected when the ant is located exactly at the nest and when in unfamiliar surroundings. We recently verified this prediction with ants tethered on an air suspended treadmill [93].

- Ants dragging a heavy food item backwards are aligned in the direction opposite to their direction of movement [94–96]. The existence of views with opposite valance learnt in the anti-goal direction may thus be recognised as repulsive and favour further backward motion when the ant body is facing away from the goal. Recent work seems to verify this prediction [97].

- Nest-oriented and anti-nest-oriented memories could be used to navigate along both inbound and outbound paths by swapping their respective valences. To achieve this, antagonist MBONs pathways could converge onto two different 'opponent integrator neurons' with opposite signs. Motivation for homing or foraging, which is believed to select a given memory bank [98–101], could instead select the 'opponent integrator neuron' with the correct sign. This could be tested behaviourally, by looking at whether homing ants released on its outbound route tend to be repulsed from the outbound direction.

## Conclusion

We adapted the well-supported idea that insects continuously combine appetitive and aversive memory pathways [63] to the context of ant visual navigation by assuming that ants learn both attractive and repulsive views, when looking in opposite direction [48]. We showed that this effectively results in a process analogous to an opponent process, where the resulting signal is the relative difference between the two 'directionally opponent' visual familiarity measurements. Contrary to a single familiarity measurement, as usually assumed, this opponent signal correlates remarkably well with the current directional error and can be used directly to drive behaviour (without the need to scan) and produce robust navigation. Other models, such as 'correspondence' models [22], can also benefit from using both aversion and attraction components, this time across the visual field [4,72]. But these models have the limitation that retinotopy must be preserved in memory, which, contrary to familiarity-based model, does not fit the mushroom body connectivity. How such an opponent signal is actually processed to drive motor behaviour in various flying or walking insect navigators remains an open question but our model suggests that robust information for directional control is already present in the Mushroom Body output neural population.

## Methods

This work aims at exposing the concept of how opponent-like processes in visual memories might be used for navigational tasks. To do so, we used a simple agent-based model in a closed loop with a 3D virtual environment. All following simulations were performed in Python 3.8.

### Virtual world and acquisition of views

The virtual environment used in our model was generated by the software Habitat3D [102], an open-source tool which provides photorealistic meshes of natural scenes from point clouds acquired with help of a LiDAR scanner (IMAGER 5006i). This environment is mapped on the habitat of *Myrmecia* ants from Canberra, Australia [69]. The mesh spans a diameter of around 65 metres and features large eucalyptus trees and distant panorama cues. This dataset can be found on the Insect Vision webpage (https://insectvision.dlr.de/3d-reconstruction-tools/ habitat3d). For speed optimization purposes, we down-sampled the originally high-resolution mesh with the open-source software Blender into a lower number of vertices; the rendering was then realized in OpenGL, with the Python libraries Plyfile and PyOpenGL.

This 3D model enabled us to render panoramic views from any location as a 360-degree picture. We chose input only to be the blue channel of the RGB scene, resulting in only one luminance value per pixel. Also, the skybox was made a pure blue uniform colour. That way, as with UV in natural scenes [103,104], blue provides a strong contrast between the sky and the terrestrial scene. This approximates the type of visual information used by navigating ants [105,106].

The rendered views were down-sampled to 72×27 px, (5°/pixel horizontal resolution) to roughly match the ant's eye resolution. Note that in one section we explored the effect of resolution (see "Parameters analysis"). Finally, these views were cropped vertically so that the bottom, floor-facing part was discarded. As a result, the visual information that the model receives is a small rectangular matrix of single-channel values representing the above-horizon panorama.

### Memorised views and current familiarity

The agent is assumed to have stored a collection of memorised views around the nest and along a route (Fig 1A). During tests, the agent computes a value of visual familiarity at each time step by comparing its current view to its memory bank. This is achieved by calculating the global root mean square pixel difference [30] between the current view and each of the views in the memory bank and keeping the value of the lowest mismatch, as typically achieved in ant navigation studies [35,36,38,50,71,72,107], models [24,27,28,32,69] and inspired robotics [22,25,108]. This mismatch value provides an estimation of the unfamiliarity of the current view. We normalise unfamiliarity values to fit the range [0:1], and then subtract the obtained value to 1 in order to get what we thus call "familiarity". Note however that the views are not rotated but compared only according to the facing directions of the current and memorised views.

### Combining attractive and repulsive visual memories

The novelty of this model is that the agent is assumed to have two independent memory banks: one 'attractive', and one 'repulsive'. The attractive memory bank includes 'learning walks views' memorised around the nest while pointing nest-ward, as well as 'homing route views' memorised along a straight, homebound route (Fig 1A). The repulsive memory bank

includes 'learning walks views' memorised around the nest while pointing out-ward (away from the nest), and possibly 'outbound route' views (see Discussion).

For simplicity, N = 25 learning walk views were assumed to be sampled along a 2m radius spiral around the nest and the homing route was chosen to be 10 metres long, fitting roughly with observed data in *Myrmecia* ants nesting in this environment [48]. We also ran simulations with other learning walk spans: 0.5, 2 and 8 metres (see S1 Fig).

At each time step, the agent computes two values of familiarity: one by comparing its current view to the attractive memory bank and one by comparing this same current view to the repulsive memory bank. These two familiarity values are assumed to have an antagonist effect on the ant's behaviour, their integration resulting in what we call here 'Opponent familiarity'. We modelled this by a simple subtraction:

$$Opponent\ familiarity = (Attractive\ familiarity\ value - Repulsive\ familiarity\ value) \qquad \text{Eq 1}$$

This opponent familiarity is what constitutes our agent's overall drive, in the sense that it modulates the turning amplitude.

$$Overall\ drive = Opponent\ familiarity \qquad \text{Eq 2}$$

For example, positive Opponent familiarity values (for 'attractive familiarity' > 'repulsive familiarity') cause the agent's current direction to be rather aligned with an attractive memorised view.

In situations where using only the attractive visual memories, the attractive familiarity value obtained was directly converted into the 'overall drive' (see below).

## Using attractive views only

We tested the agent using the attractive memory bank only. In such cases, the Attractive familiarity value was used directly to generate the overall drive. To facilitate the comparison with the use of opponent visual memories (see previous section), we normalised the Attractive familiarity by subtracting an 'average world familiarity value' corresponding to the average familiarity obtained between two views in the world, estimated by comparing 32 views at random positions and orientation.

$$Overall\ drive = Attractive\ familiarity\ value - average\ world\ familiarity\ value \qquad \text{Eq 3}$$

That way, the Overall drive is normalised around 0, as when combining Attractive and Repulsive memories. Positive values indicate a rather good match (high familiarity) while negative values indicate a rather poor match (low familiarity). Note that we performed a systematic parameter analysis that enabled us to pinpoint the best set of parameters (in that case, varying the 'baseline' parameter (see next section) amounted to varying this 'average familiarity value'). In other words, the rather arbitrary choice of the 'average world familiarity value' had no incidence on the following analysis.

## Oscillatory agent

To drive the agent, we used a similar approach to the ones of Kodzhabashev and Mangan [25] and Wystrach et al., [80]. The agent is a simple dot in space (x, y) with a current heading (theta). The elevation of the agent (z) was fixed at a height of 2 centimetres above the ground. The agent has a continuously running oscillator alternating between left mode and right mode, which controls the instantaneous turning direction: the 'Turn direction' thus alternates at each time step (Left-Right-Left-Right- . . .) and is solely controlled by the oscillator. However, the 'Turn amplitude' is directly dependent on the current overall drive (see previous

section), that is, on the current visual familiarities.

$$Turn\ amplitude\ (deg) = baseline - (gain \times overall\ drive) \qquad \text{Eq 4}$$

We used a single parameter (gain) to convert the overall drive into the angular value for the turn amplitude. The baseline was a fixed parameter which controlled the extent of the turn amplitude given an *Overall drive = 0*. Therefore, a positive overall drive will reduce the current turn amplitude from baseline towards 0 degrees (straight ahead), while a negative overall drive will increase the current turn amplitude from baseline towards 180 degrees (opposite direction). The turn amplitude was clipped between 0 and 180 degrees.

Across time steps (t), the agent's orientation (theta) will thus oscillate between left and right turns ($(-1)^t$). So, to update the current facing direction of the agent:

$$Theta_{(t+1)} = Theta_{(t)} + (Turn\ amplitude \times (-1)^t) + noise \qquad \text{Eq 5}$$

At each time step, we added noise as a random angular value drawn from a Gaussian (*mu = 0*; *std = 10 degrees*). This was to ensure our model is robust to the intrinsic noise of the real world.

Once released at a given location, the agent repeats the current rules at each time step:

1. Obtain the current view and compute its corresponding overall drive (Eqs 1 and 2 or Eq 3)

2. Turn on the spot (turn amplitude determined by Eq 4, turn direction determined by the oscillator state: Eq 5)

3. Walk 1 step forward (we set 1 step = 20 cm)

It is useful to note that the turn performed in rule 2 represents the direction in which the forward step will be performed, and not a turn performed independently from the forward motion as in 'stop and scan' models.

We released the agent equipped with its Attractive and Repulsive memory banks (or Attractive only memory bank) at different release points, and let it run in a closed loop with the environment, and observed the resulting paths (Figs 3 and 4).

## Sampling of the apparent familiarity

To analyse the differences between direct (Attractive only) and opponent (Attractive–Repulsive) memories, we performed a somewhat exhaustive sampling of the apparent visual familiarity around the nest: we defined a lattice of 3600 sampling positions, covering a square region of 30 × 30 metres centred on the nest, every 50 cm. On each of these positions, an agent was positioned and acquired a reading of the visual familiarity in all the directions (72 positions all around, i.e. every 5 degrees). We thus obtained maps of the familiarity in this area, for both memory types (Fig 2).

## Parameter analysis

We performed a systematic parameter analysis to determine the range of gain and baseline in which the agent is able to operate. For each combination of parameters, *M* individual runs of 320 steps (enabling the agent to walk 320 x 0.2 = 64 meters) were performed from *M* different release points, equally spaced around the nest *(M = 10*, i.e. one every 36 degrees around the nest). For each of these individual runs, the initial walking direction was randomly picked. The following parameters were varied:

1. Release distance: The release distance was varied between 0 and 32 metres away from the nest, which correspond to the limits of our reconstructed world. Most agents were unable to home when released at 32 metres.

2. Learning walk span: we ran parameter exploration simulations with three different learning walk spans: 0.5, 2 and 8 metres (S1 Fig).

3. Oscillator baseline: As described in the 'Oscillating agent' section, the oscillator driving the turning direction has a baseline firing value. We varied this baseline between 0 degrees (= no spontaneous oscillations, the agent can only up-regulate the turn amplitude when the view is rather repulsive) and 180 degrees (= spontaneous full U-turn, the agent can only down-regulate turn amplitude when the view is rather attractive).

4. Gain: The turning amplitude is governed by the gain as described in the 'Oscillating agent' section. We varied this gain parameter between 0 and +inf.

The success of each parameter combination was determined by taking the median arrival distance (i.e., distance from the nest after the 320 steps) of the 10 individual agents. We chose the median as it provides a good estimation as to whether the majority of agents could reach the nest, while the mean would be very sensitive to the paths effected by lost agents.

Finally, we also investigated the robustness of using opponent process memories by adding the following constraints:

- Directional noise during learning: the angle at which the learning walk views were taken was submitted to angular noise (up to 90 degrees in both directions, randomly taken from a uniform distribution)

- Attractive/Repulsive memories decoupling: the positions at which Attractive and Repulsive view were taken were made uncorrelated, by using two different learning walk spirals.

## Supporting information

**S1 Fig. Effect of the learning walk span.** Heatmaps of the interaction between the gain and baseline parameters, and their effect on homing success (distance of arrival at the end of the simulation), for two sizes of learning walks and three release distances.
(TIFF)

**S2 Fig. Virtual 3D worlds.** Snapshots of the 3D point clouds of the two virtual environments used in this work. Top, Canberra environment, large scale and large distant features such as trees; Bottom, Sevilla environment, with high clutter and no distal panorama.
(TIFF)

## Acknowledgments

We thank Jochen Zeil and Trevor Murray for fruitful discussions about the model's principles, as well as Sebastian Schwarz and Cody Freas for comments on a previous version of the manuscript.

## Author Contributions

**Conceptualization:** Antoine Wystrach.

**Formal analysis:** Florent Le Möel, Antoine Wystrach.

**Funding acquisition:** Antoine Wystrach.

**Investigation:** Florent Le Möel, Antoine Wystrach.

**Methodology:** Florent Le Möel, Antoine Wystrach.

**Resources:** Antoine Wystrach.

**Software:** Florent Le Möel, Antoine Wystrach.

**Supervision:** Antoine Wystrach.

**Writing – original draft:** Florent Le Möel, Antoine Wystrach.

**Writing – review & editing:** Florent Le Möel, Antoine Wystrach.

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
