## [Decision Letter · Decision Letter 0]

23 Sep 2019

Dear Dr Wystrach,

Thank you very much for submitting your manuscript 'Opponent processes in visual memories: a model of attraction and repulsion in navigating insects' for review by PLOS Computational Biology. Your manuscript has been fully evaluated by the PLOS Computational Biology editorial team and in this case also by independent peer reviewers. The reviewers appreciated the attention to an important problem, but raised some substantial concerns about the manuscript as it currently stands. While your manuscript cannot be accepted in its present form, we are willing to consider a revised version in which the issues raised by the reviewers have been adequately addressed. We cannot, of course, promise publication at that time.

Sincerely,

Joseph Ayers, PhD

Associate Editor

PLOS Computational Biology

Wolfgang Einhäuser

Deputy Editor

PLOS Computational Biology

[LINK]

Reviewer's Responses to Questions

**Comments to the Authors:**

Reviewer #1: This manuscript proposes the novel idea that central foraging insects learn and use ‘opponent’ visual memories to navigate. The core idea of ‘repulsive’ visual memories is intrinsically interesting. Yet, my main concern is that it is partly motivated by inaccurate interpretation of former works and hasty claims. In particular, the fundamental drawback of the current models, as criticized by the authors, exists only if one assumes that insects can recognize a view only when properly aligned. This hypothesis might be supported by some data, but remains to be clearly demonstrated. Thus, “current models” indeed do not satisfy or even mention this precondition.

Another claim is that the present study would propose a likely neural implementation. I am not convinced that the cartoon of a putative single inhibitory synapse (Fig. 1C) and mere parallel with Drosophila’s olfactory and appetitive neural system (l. 395-423) are sufficient to qualify as a likely neural implementation. Likewise, as most predictions (l. 425-453) rely on this far-fetched neural implementation, they sound more like long-term future research axes than actual experimentally-testable predictions.

As this work explores (but does not systematically assess) a new concept to model a behavior – insect visual navigation – of which most underlying neural mechanisms, as well as the exact nature of the visual cues involved, are still unknown, I would recommend to submit a new version with more cautious claims and balanced argumentation. Separate results and discussion sections would also make things clearer.

Major comments:

The authors begin their argumentation by describing what they call “a widespread orthodoxy” (l. 73) which they define by the consecutive statements (l. 73-80):

(i) “insects memorise these scenes in an egocentric way”

(ii) “the recognition of a memorised scene … is dependent on the insect’s current head direction”

(iii) “current view will appear familiar only when the insect is aligned in roughly the same direction that it was at the time of learning. If the insect is located at a familiar location, but its body is oriented in any other direction, its current view would be different and the location would appear unfamiliar.”

Although (i), (ii) and (iii) are articulated by the authors as equivalent assertions, they are not in my opinion. (i) seems too strong as some authors would claim that all navigational information is combined on a geocentric map, and others have proposed that snapshots would be memorized with the celestial compass direction to allow, during later recall, proper alignment and comparison with the current view. (ii) is what makes such image-based homing possible. Even if mental rotation was used (see below), the amount of rotation required would depend on the current insect heading, and hence would allow for proper correction of the current heading. (iii) is not valid in the case of mental rotation (see below). Also, it seems unlikely for a lost ant to ‘hit’ a familiar route under the correct angle, and thus searching would be particularly ineffective.

Then it is claimed that current models solve that issue by a stop-scan-go strategy, that is condemned as being “cumbersome”, “non-parsimonious” and (biologically) “unrealistic”. This might stem from misunderstanding the models based on computing rotational image difference functions (RIDFs). As in (50, 51, 56), multiple RIDFs are computed from rotating the current view against the multiple memorized snapshots, and the direction to steer is then given by a weighted circular mean of the directions at the RIDF minima.

However, physical rotation is not the only way to compute RIDFs. As discussed in the attached manuscript under review, mental rotation is also possible and has been suggested already by several authors. In fact, RIDF-based models like (51) leave open how RIDFs are computed. In principle, to find maximum-match directions, RIDFs could be sampled by physical and/or mental rotations, covering fully or partially 360°, in a systematic or a random way. A typical RIDF profile shows a more or less narrow and deep minimum difference (i.e. max familiarity) when both images align to the same (goal) direction, and plateaus at higher values elsewhere. In the present model, the combined ‘opponent’ RIDF (Fig. 1B) also show a maximum difference (i.e. min familiarity) in the anti-goal direction. The ‘opponent’ RIDFs thus present steeper gradients than the flatter profiles of standard RIDFs, hence, helping any algorithm to find peaks. This, I think, is what the authors implied with “recovering the correct direction through scanning … [by] increase[ing] the difference between incorrect and correct directions” (l. 241-242), albeit in a less clear way to me.

In the present manuscript, a single point of ‘opponent’ RIDF is sampled at each time step using oscillatory and noisy physical rotations. Proof of principle is given that this strategy is effective when the agent turns with an amplitude modulated by the sampled value, but I am not convinced that this approach outclasses the other models with respect to the three criteria mentioned by the authors:

Cumbersomeness:

Unlike physical 360°-scans on the spot, mental rotation is not cumbersome as it could be done “en route”.

Parsimony:

Although performing systematic mental rotation continually may appear computationally “non-parsimonious” compared to physical rotation, learning goal-directed and anti-goal views seems somewhat superfluous for animals with nearly 360° field of view. Why would they have to turn back to learn the already visible opposite view? Can’t they learn both views at once? Moreover, anti-goal views seem somewhat redundant with goal-directed views acquired while spiraling around a place. In fact, goal-directed views already cover “opponent” directions that, like in (51), can compete to steer the animal. Admittedly, the strategy proposed by the authors is parsimonious in that it computes only one RIDF value at a time, but the agent can then only determine how much to turn, not in which direction to turn. Hence, it does “waste” information contained in the snapshot memories, namely the proper direction to steer. That is, the proposed approach is parsimonious in computation but not in memory.

Biological plausibility:

Stop-scan-go strategy is not that biologically “unrealistic” as it was reported for ants released at unfamiliar locations (28, 54). Mental rotation is not “unrealistic” either, as a recent neural ring model (10), using a clockwise and counterclockwise-shifted neural mapping, can compare whether a left or right rotation would produce a better match between the current compass input and the path-integration memory. Furthermore, the authors generalize their opponent-memory strategy to route navigation. As no “look-back” during route learning has been reported, they argue (especially in their other manuscript under review) for the use of outbound memories as opponent views during inbound journeys. However, this does not fit well with trained individuals following distinct inbound and outbound paths, to and from the same feeder (32). Intuitively, if ants were simply inverting attracting/repelling roles among the same set of visual memories, one would expect that each individual would eventually travel on a unique bidirectional path (per feeder).

Minor comments:

l. 149-150: “we implemented” but ref. 44 is not authored by any author of the present manuscript

l. 248-279: The present model computes a difference, not a ratio. A difference is expected to scale with its operands, thus these paragraphs are not that clear to me.

l. 451-452: Motivation-based selection of memory banks has also been proposed and tested in:

- Hoinville T, Wehner R, Cruse H. Learning and retrieval of memory elements in a navigation task. In: Biomimetic and Biohybrid Systems. Springer; 2012; pp. 120-131.

- Cruse H, Wehner R. (2011) No need for a cognitive map: decentralized memory for insect navigation. PLoS computational biology, 7(3), e1002009.

l. 512-513: The definition of familiarity as the inverse of RMS pixel difference seems dubious as it is likely to explode to overly large values or infinity when views would match closely. Also, it does not fit with the range [0:1] displayed on the figures.

l. 595-599: Although this algorithmic formalization is fine, it might confuse some readers with the ‘stop-scan-go’ strategy criticized earlier in the text. The critical point here is that there is no ‘stop’ step, and the ‘turn on the spot’ step is not a scanning procedure (e.g. sampling RIDFs), but simply an action. I think an explicit note would help to clarify that.

Fig. 1B: Although familiarity is function of direction, using xy-plots instead of polar plots could be more readable and help to relate to standard RIDF plots (albeit familiarity is inverse of difference). Also, polar plots tend to compress the middle and enlarge the sides. Thus, the two peaks of the ‘integration’ curve are equivalent, yet the peak at -1 is shrunk compared to the other at +1.

Fig. 2A: The arrows are too little to be seen. I am not sure what “directionality” refers to and if it is connected to the “specificity” described in the text.

Fig. 2B: Choosing a viewing direction perpendicularly to the learned route is peculiar, why not directed along the route? How is calculated the familiarity between the memory bank and the current view? In other words, how one value is drawn from multiple familiarity values? The horizontal panels below are confusing as they represent vertical transects (especially when considered the horizontal route).

Fig. 2C: “Deviation” may be preferred to “angular distance”.

Fig. 3 & 4: Quality of plots is poor, please avoid low-resolution bitmaps.

Fig. 3A,B: Giving just single examples for ‘decoupled memories’ and ‘noisy learning angle’ scenarios is unconvincing to assess the robustness of the model to these critical parameters.

Reviewer #2: This model makes a very valuable contribution to understanding insect navigation. In particular, it brings into focus the potential importance of aversive mechanisms, even for performing behaviours that intuitively appear to be attractive. I think this will have a profound effect on our understanding of navigation, regardless of how accurate any of the details of the model turn out to be in the future.

The model builds on a family of models developed over the last 10-15 years to implement visual route following using a holistic familiarity encoding. This type of model has the great appeal that is simple and is consistent with what is known about insect mushroom bodies, where such memories are likely to be encoded. Previous models of this sort have simply encoded a set of views towards the nest as attractive. The novel idea here is also to encode a set of views away from the route as aversive. The authors show that by combining both sets of views, an agent can navigate far more reliably. They show a variety of ways in which the robustness of homing is significantly improved. The effects are impressive.

The idea is implemented with a previously published oscillatory model of navigation that works effectively by turning away more from less attractive views. In previous models, the level of attractiveness has been determined simply from a familiarity measure. The novel idea here is that attractiveness is the difference between familiarity in the nest-ward (stored as an ‘attractive memory bank’) and familiarity in the anti-nest-ward direction (stored as an ‘aversive memory bank’). The authors’ novel idea of using both an attractive and an aversive memory bank could potentially also be usefully adapted for other types of models.

Suggestions:

In the Introduction, it might be useful to make it clearer that there are many classes of models for visual homing in insects, and that your discussion of limitations apply to the class that your model comes out of.

In the Discussion, it might be helpful to note that with this new addition, this school of model can now do some of what a different class of ‘correspondence’ models (which unlike the present models require retinotopy to be preserved in the encodings) can do - namely home robustly from away from a route corridor. It is also interesting to note what this new model shares with the first computational model proposed by Cartwright and Collett. Although the two models are different in many key respects, they share the finding that guidance based on attraction alone is not robust, while one using a combination of attraction (to views indicating the nest is in front) and aversion (to views indicating the nest is behind) is robust.

As a general note about the writing style, try to avoid starting each paragraph by referring directly to the previous paragraph (e.g. ‘this problem’, ‘this hypothesis’, ‘in such a way’, ’Moreover’, ’However’). Certainly do not start new sections ‘in such a way’. It is much easier for readers if paragraphs, and particularly sections, are self contained.

Line 112: “Limits of the current models.” It needs to be clear that you are referring to only a subset of the current models, as not all models have the limits your refer to. See also comment to line 129.

Line 119: insert “It is problematic [for these models] to infer … “

Line 120: insert “In several [of these types] of models … “

Line 129: “We present here a simple solution to this problem. … “ Referring back to the previous section, I think ‘this problem’ refers to the need to sample several directions in order to choose the best. In the implementation here, this problem is solved by using an oscillatory path and continuously modulating the amount of turning. From my reading, using the two memory banks solves a different problem that is addressed later: i.e. robustness. It might be helpful for the reader if you separated the two issues. Moreover, given that you are using an oscillatory model that is similar to previously published ones, it is perhaps not helpful to have the need for scanning in the ‘limits of the current models’, since (as I understand it - or possibly mis-understand it) you are not providing a new solution to that.

Figure 2: There may well be arrows in this figure, but they’re not visible with the resolution of the pdf that I have. It would probably be worth making the arrows larger. Possibly that might require making the grid cells larger too.

Line 214: This section should be called something like “The drawbacks of using only goal-oriented views [without scanning]”

You are imposing the constraint of facing in only one direction. So in line 216, change ‘while’ to ‘if’.

Line 221: insert “… [without scanning], familiarity does not reliable indicate … “

Line 229: As an example to my general point about writing style, start instead with something like “We show here that [the need for scanning] can vanish … “

Figure 3: It is clear from this figure that the addition of the repulsive visual memories makes homing much more robust. But I find the details quite confusing, and in particular I don’t understand the pattern of failures for the attractive views only. What I understand, after spending a while trying to make sense of it, is that ants are trained at the grey dots, facing towards the nest, and along a nest-ward route indicated by the grey line (??). In this case, it makes some sense to me that the ants can home from along the route, and that they are not good at homing from the opposite side of the nest. But it puzzles me why the trajectories on the opposite side of the nest appear to go nowhere at all. Is that because of the choice of the ‘optimal parameters’? Are optimal parameters chosen for each release site, and is optimality based on end distance from the nest? Thus choosing not moving about moving in the wrong direction?

I also don’t understand the difference between A and B. Are they trained at the same location but tested at different locations? The positions of the release sites look the same so I don’t think that can be it. Or is the entire experiment, including both training and testing, shifted to a new location for B? This would explain where there is some homing in the right hand panel of B. But then I don’t understand why there would be no homing from the route (if the grey line is indeed the route) in the left hand panel of B. What is it about the parameters and different location that make so many trajectories head off towards the NE?

If the parameters are so critical to the success of using attractive views only, please try to describe what is needed more fully. Why some parameters work, why others don’t, and why the difference when shifting to a new location. Also, why was there no problem in getting homing in references 28 and 41 referred to in the Introduction?

Figure 4 shows nicely how robust the opponent-based navigation is.

C. insert: “half [as many] memories”

D. [Lower] resolution

In many of the panels in Figures 3&4 there is something special about the SW-NE direction. Why?

Lines 444-448: This prediction appears to be counter to the findings of Schwarz et al 2017, in which the same authors found that using route memories to travel backwards required ‘peeking’ along the route to set the direction. Can you reconcile the previous findings with the current prediction?

**Have all data underlying the figures and results presented in the manuscript been provided?**

Reviewer #1: No:

Reviewer #2: Yes

PLOS authors have the option to publish the peer review history of their article (what does this mean?). If published, this will include your full peer review and any attached files.

Reviewer #1: No

Reviewer #2: No

---

## [Decision Letter · Decision Letter 1]

10 Dec 2019

Dear Dr Wystrach,

Thank you very much for submitting your manuscript, 'Opponent processes in visual memories: a model of attraction and repulsion in navigating insects', to PLOS Computational Biology. As with all papers submitted to the journal, yours was fully evaluated by the PLOS Computational Biology editorial team, and in this case, by independent peer reviewers. The reviewers appreciated the attention to an important topic but identified some aspects of the manuscript that should be improved.

We would therefore like to ask you to modify the manuscript according to the review recommendations before we can consider your manuscript for acceptance. Your revisions should address the specific points made by each reviewer and we encourage you to respond to particular issues Please note while forming your response, if your article is accepted, you may have the opportunity to make the peer review history publicly available. The record will include editor decision letters (with reviews) and your responses to reviewer comments. If eligible, we will contact you to opt in or out.raised.

- Supporting Information uploaded as separate files, titled 'Dataset', 'Figure', 'Table', 'Text', 'Protocol', 'Audio', or 'Video'.

We hope to receive your revised manuscript within the next 30 days. If you anticipate any delay in its return, we ask that you let us know the expected resubmission date by email at ploscompbiol@plos.org.

Sincerely,

Joseph Ayers, PhD

Associate Editor

PLOS Computational Biology

Wolfgang Einhäuser

Deputy Editor

PLOS Computational Biology

[LINK]

Reviewer's Responses to Questions

**Comments to the Authors:**

Reviewer #1: Thanks for your edits. Focusing explicitly on a specific class of models now sounds fairer. However, “familiarity-based models” seems inaccurate. “Image rotation” or more precisely “Rotational Image Difference Functions” (RIDFs) is a central concept (although never mentioned in the paper) to several familiarity-based models. In several cases, e.g. ref. 28, the model is agnostic as to how RIDFs are computed. It could be done by on-the-spot scans as criticized in the paper, or it could be done by some mental rotation (also not mentioned in the paper) as I already objected. Perhaps, the authors do not see how mental rotation could be neurally implemented, and it seems indeed less parsimonious (neurally) than a simple image difference, but to my knowledge there is yet no evidence excluding mental image rotations. It would be more accurate to call the addressed class of models “stop-scan-go models” and describe them as a certain way, namely embodied image rotation, to implement familiarity/RIDFs-based models.

To defend that they would propose a “likely neural implementation”, the authors rely on the following reference-supported arguments: mushroom bodies (MBs) are well conserved across species; MBs process olfactory and visual inputs; MBs project excitatory and inhibitory outputs; and MBs are crucial for appetitive and navigation behaviors. But why, within MBs, the olfactory circuits would indicate anything about the visual circuits, furthermore when considered in two species subject to quite different contexts, walking vs. flying, foraging centrally vs. non centrally, social vs. non social? Inhibitory and excitatory neurons are common to quite many brain areas, including MBs. Using that as supporting that MBs store views in an opponent way appears far-fetched to me.

Despite those considerations, I recommend this paper for publication.

Reviewer #2: This manuscript is now much improved and I have only a few minor comments. As before, I think that the model has the potential to have a big impact on how we think about navigation in insects, and so is well worth publishing here.

line 29: compares

line 126-7: “retinotopy does not need to be preserved for the comparison stage”. I’m not sure this is necessarily correct. I think you mean that you don’t need a retinotopic mapping.

line 145: “some drawbacks”. this paragraph seems to be about one drawback.

line 169:”the problem of familiarity-based models”. You seem to be saying here that familiarity-based models are a problem in themselves - which is obviously not what you mean.

Line 173-4. Since you make such a big thing of it in the next sentence, this particular assumption that both outputs are continuously integrated does appear to be made through the desire to improve a model rather than flowing from observations.

line 184: “this hypothesis” is that the solution you present in the previous paragraph is correct?

Lines 220-270 appear to be more or less repeating lines 90-167. Can they be omitted or merged into the existing Introduction?

line 277: subtracted from

line 281: omit ‘rather’

line 356: ‘the’ instead of ‘this’

**Have all data underlying the figures and results presented in the manuscript been provided?**

Reviewer #1: Yes

Reviewer #2: Yes

PLOS authors have the option to publish the peer review history of their article (what does this mean?). If published, this will include your full peer review and any attached files.

Reviewer #1: No

Reviewer #2: No

---

## [Decision Letter · Decision Letter 2]

4 Jan 2020

Dear Dr Wystrach,

We are pleased to inform you that your manuscript 'Opponent processes in visual memories: a model of attraction and repulsion in navigating insects' has been provisionally accepted for publication in PLOS Computational Biology.

In the meantime, please log into Editorial Manager at https://www.editorialmanager.com/pcompbiol/, click the "Update My Information" link at the top of the page, and update your user information to ensure an efficient production and billing process.

One of the goals of PLOS is to make science accessible to educators and the public. PLOS staff issue occasional press releases and make early versions of PLOS Computational Biology articles available to science writers and journalists. PLOS staff also collaborate with Communication and Public Information Offices and would be happy to work with the relevant people at your institution or funding agency. If your institution or funding agency is interested in promoting your findings, please ask them to coordinate their releases with PLOS (contact ploscompbiol@plos.org).

Thank you again for supporting Open Access publishing. We look forward to publishing your paper in PLOS Computational Biology.

Sincerely,

Joseph Ayers, PhD

Associate Editor

PLOS Computational Biology

Wolfgang Einhäuser

Deputy Editor

PLOS Computational Biology

Reviewer's Responses to Questions

**Comments to the Authors:**

Reviewer #1: Thanks for your answers and happy new year!

**Have all data underlying the figures and results presented in the manuscript been provided?**

Reviewer #1: Yes

PLOS authors have the option to publish the peer review history of their article (what does this mean?). If published, this will include your full peer review and any attached files.

Reviewer #1: No

---

## [Editor Report · Acceptance letter]

29 Jan 2020

PCOMPBIOL-D-19-01236R2 

Opponent processes in visual memories: A model of attraction and repulsion in navigating insects’ mushroom bodies

Dear Dr Wystrach,

I am pleased to inform you that your manuscript has been formally accepted for publication in PLOS Computational Biology. Your manuscript is now with our production department and you will be notified of the publication date in due course.

With kind regards,

Sarah Hammond
